# Random Regression Model for Genetic Evaluation and Early Selection in the Iranian Holstein Population

**DOI:** 10.3390/ani11123492

**Published:** 2021-12-07

**Authors:** Yasamin Salimiyekta, Rasoul Vaez-Torshizi, Mokhtar Ali Abbasi, Nasser Emmamjome-Kashan, Mehdi Amin-Afshar, Xiangyu Guo, Just Jensen

**Affiliations:** 1Department of Animal Science, Science and Research Branch, Islamic Azad University, Tehran 14515-775, Iran; yasamin.sa@qgg.au.dk (Y.S.); nasser_ejk@yahoo.com (N.E.-K.); aminafshar@gmail.com (M.A.-A.); 2Department of Animal Science, Faculty of Agriculture, Tarbiat Modares University, Tehran 14115-111, Iran; 3Department of Animal Breeding and Genetics, Animal Science Research Institute of Iran, Karaj 31585-963, Iran; pmaz_abbasi@yahoo.com; 4Center for Quantitative Genetics and Genomics, Aarhus University, 8830 Tjele, Denmark; xiangyu.guo@qgg.au.dk (X.G.); just.jensen@qgg.au.dk (J.J.)

**Keywords:** early sire selection, cross-validation, genetic evaluation, test-day model

## Abstract

**Simple Summary:**

The objective of this study was to use a model to predict breeding values for sires and cows at an early stage of the first lactation of cows and progeny groups in the Iranian Holstein population to support the early selection of sires. Our results show that we can select sires according to their daughters’ early lactation performance before they finish first lactation. Cross-validation results show that early selection accuracy can be high, and such an early selection can decrease the generation interval and lead to an increased genetic gain in the Iranian Holstein population.

**Abstract:**

The objective of this study was to use a model to predict breeding values for sires and cows at an early stage of the first lactation of cows and progeny groups in the Iranian Holstein population to enable the early selection of sires. An additional objective was to estimate genetic and phenotypic parameters associated with this model. The accuracy of predicted breeding values was investigated using cross-validation based on sequential genetic evaluations emulating yearly evaluation runs. The data consisted of 2,166,925 test-day records from 456,712 cows calving between 1990 and 2015. (Co)-variance components and breeding values were estimated using a random regression test-day model and the average information (AI) restricted maximum likelihood method (REML). Legendre polynomial functions of order three were chosen to fit the additive genetic and permanent environmental effects, and a homogeneous residual variance was assumed throughout lactation. The lowest heritability of daily milk yield was estimated to be just under 0.14 in early lactation, and the highest heritability of daily milk yield was estimated to be 0.18 in mid-lactation. Cross-validation showed a highly positive correlation of predicted breeding values between consecutive yearly evaluations for both cows and sires. Correlation between predicted breeding values based only on records of early lactation (5–90 days) and records including late lactation (181–305 days) were 0.77–0.87 for cows and 0.81–0.94 for sires. These results show that we can select sires according to their daughters’ early lactation information before they finish the first lactation. This can be used to decrease generation interval and to increase genetic gain in the Iranian Holstein population.

## 1. Introduction

Genetic progress in dairy cattle strongly depends on the merit of bulls used as sires for the next generation. Therefore, selecting sires is a vital decision that affects future production, health, and profit in the following generations of dairy cows [1]. The merit of selected sires is affected by factors such as the pedigree merit of parents, the number of bulls sampled, the speed and accuracy of progeny testing, the intensity of selection following the test, and the maximum use of the best retained bulls [2]. The population accuracy of predicted breeding values (EBV) of candidates is a critical point in selection programs. Besides, a higher population accuracy of selection candidates connotes genetic progress [3,4]. The traditional selection of dairy bulls is based on pedigree selection and progeny testing. Efficient selection includes recognizing young bulls of high genetic merit using pedigree information [5]. In progeny testing, the genetic evaluation of bulls is based on the performance of offspring [6]. In more developed countries, the selection of dairy bulls is based on genotyped information from dense SNP arrays to predict genomic EBV using genomic prediction procedures. In addition, this information can be used for accurate parentage assignment and pedigree reconstruction [7]. Therefore, genomic selection is replacing the traditional selection based on pedigree [7,8,9].

Although the evaluation of sires can be more accurate when including later records (e.g., second and third lactations) compared with early daughter records, selecting according to later records may have some disadvantages. It can be biased by selection and low production caused by abortion or health troubles [9]. In addition, recognizing superior sires as soon as possible (according to their daughters’ early records of lactation) results in selecting sires sooner than when waiting for their daughters to finish their first lactation. This early selection leads to a decrease in generation interval, and thus speeds up genetic changes in the population, which is profitable for the dairy industry [1].

In many previous studies, animal and sire models were used to estimate genetic parameters of milk yield [10,11,12,13]. Milk yield is highly affected by the stage of lactation expressed as different days in milk (*DIM*). Since genetic parameters change over the time, an assumption of flexible genetic parameters across *DIM* is essential. Besides, variance components change during time, mainly because of the genetic selection [8,14]. Random regression (RR) models are often used for the genetic evaluation of traits that are measured at different times during the lifetime or physiological cycle of an animal because it allows studying the variation of a trait as a function of time (age; days in milk) [15,16]. 

Cross-validation of evaluation models is based on a series of statistics, including the change of predictions from older to more recent evaluations that can be used to evaluate the accuracy of an early prediction of breeding values [17]. The classical accuracy of EBV is defined as the correlation between the true breeding value (TBV) and EBV for individuals across repeated sampling and is a measure of potential change in EBV based on increasing amounts of information [18]. Cross-validation methods can be used to evaluate population accuracies of predicted breeding values and potential biases in predictions [19,20]. Cross-validation is an efficient tool for validating RR models since it can include partial data on lactations in progress [20].

The objectives of this study were: (1) use the RR model for genetic evaluation in the Iranian Holstein population and to estimate genetic and phenotypic parameters associated with this model, and (2) validate a model that can predict breeding values for sires and cows at an early stage before all cows have finished their first lactation, to support the early selection of sires.

## 2. Materials and Methods

### 2.1. Data

Data on milk production test-day records of first parity of Iranian Holstein cows freshening from 1990 to 2015 were obtained from the Animal Breeding Center of Iran. Data editing was completed with own R scripts [21]. Only cows with test-day milk yield in the range 5–60 kg, were included in the analysis. Test-day records before day 5 and after day 305 were removed. Only cows with three milkings per day were selected, as this is the standard in large Iranian dairy herds. Cows were discarded if they had fewer than two test-day records. Cows, that were out of the range of 20–50 months for first calving were deleted. Herds and years, which had fewer than 500 and 1000 observations, respectively, were discarded. In addition, pedigree was traced as far back as possible. On average, five generations for animals with data traced back and the pedigree file contained 737,738 animals. The original dataset consisted of 3,042,354 records. After editing data, 2,166,925 records from 456,712 cows out of 6542 sires and 328,659 dams remained.

### 2.2. Model

Genetic parameters were estimated using a random regression model as follows: 
(1)
$$y_{ijklmn}=hy_{i}+dim_{j}+age_{k}+htd_{l}+\overset{3}{\underset{t=0}{\sum }}\emptyset_{t}(DIM_{j})\;a_{mt}\;+\overset{3}{\underset{t=0}{\sum }}\emptyset_{t}\left(DIM_{j}\right){pe}_{mt}+e_{ijklmn}$$

where 
$y_{ijklmn}$
 is observation of test-day record *n* of cow *m* obtained at days in milk (*DIM*) *j* in herd-test-day *l* in herd-year *i* of a cow calved at *k*^th^ age class; 
$hy_{i}$
 is the fixed effect of herd-year (*i* = 1,…, 7653); 
$dim_{j}$
 is the fixed effect of *DIM* (*j* = 5,…, 305); 
$age_{k}$
 is the fixed effect of age at calving (*k* = 20,…, 50 month); 
$htd_{m}$
 is the herd-test-day random effect (*m =* 1,…, 12,756); 
$a_{mt}$
 is the random additive genetic effect of *m^th^* animal; 
$\emptyset_{t}$
 is the *t^th^* coefficient of Legendre polynomials evaluated at *DIM*
*j* which is standardized between –1 and +1; 
$pe_{mt}$
 is the random permanent environmental effect; and 
$e_{ijklmno}$
 is the random residual error effect.

The distributions of random effects were assumed to be normal:
(2)
$$a\;\sim \;N\left(0,\;A⊗G_{0}\right)$$


(3)
$$pe\;\sim \;N\left(0,\;I_{q}⊗P_{0}\right)$$


(4)
$$e\;\sim \;N\left(0,\;I_{n}\sigma_{e}^{2}\right)$$


(5)
$$htd\;\sim \;N\left(0,\;I_{p}\sigma_{htd}^{2}\right)$$

where 
A
 is the relationship matrix of the order of number of animals;
$\;G_{0}$
 and 
$P_{0}$
 are matrices of the additive genetic and permanent environmental (co)-variances of random regression coefficients of the order of number of Legendre polynomials; 
$I_{q}$
 is an identity matrix of the order corresponding to permanent environmental effect which is computed for animals with record; 
$I_{n}$
 is an identity matrix of the order of records; 
$\sigma_{e}^{2}$
 is residual error variance; 
$I_{p}$
 is an identity matrix of the order corresponding to 
htd
 effect; 
$\sigma_{htd}^{2}$
 is the variance of *htd* effect; and 
⊗
 is the Kronecker product.

All analyses were performed using RR models using the average information (AI) restricted maximum likelihood (REML) module in the DMU software [22]. Legendre polynomial functions were chosen to fit the lactation curves in the RR test-day model for estimating (co)-variance components, and Legendre polynomials were generated by R script using the Orthopolynom package [23]. Various orders of polynomials were tested, and the optimum was selected based on likelihood ratio tests. The fixed parts of the curve were modeled using a step function that could move freely at every *DIM*. This was to ensure that the form of the general lactation curve was not limited by the number of parameters in the Legendre polynomials.

### 2.3. Prediction of Breeding Values 

The formulas below were used to estimate breeding value of animals on each day of lactation curve.

(6)
$${EBV}_{mj}=z_{j}{}^{\prime }{\hat{a}}_{m}$$


(7)
$$z_{j}=\left(\begin{array}{c} \emptyset_{0j}\\ \emptyset_{1j}\\ \emptyset_{2j}\\ \emptyset_{3j} \end{array}\right)\;\;\;\;\;\;\;\;\;\;\;{\hat{a}}_{m}=\left(\begin{array}{c} {\hat{a}}_{m0}\\ {\hat{a}}_{m1}\\ {\hat{a}}_{m2}\\ {\hat{a}}_{m3} \end{array}\right)$$

where 
$EBV_{mj}$
 is the estimated breeding value of animal *m* at *DIM*
*j*, 
$z_{j}$
 is a vector of Legendre polynomial coefficients evaluated at *DIM*
*j*, and 
${\hat{a}}_{m}$
 is a vector of the estimates of additive genetic random regression coefficient specific to the *m^th^* animal.

In addition, the total EBV of animal *m* was obtained by summing the EBVs from day 5 to 305:
(8)
$$EBVT_{m}=\overset{305}{\underset{j=5}{\sum }}EBV_{mj}$$


### 2.4. Estimation of Genetic Parameters

The following formulas were used for estimating additive genetic, permanent environmental variances, heritability, and repeatability of milk yield:
(9)
$$\sigma_{aj}^{2}=z_{j}{}^{\prime }G_{0}z_{j}$$


(10)
$$\sigma_{pej}^{2}=z_{j}{}^{\prime }P_{0}z_{j}$$


(11)
$$h_{j}^{2}=\sigma_{aj}^{2}/\sigma_{Phj}^{2}$$


(12)
$$r_{j}=\left(\sigma_{aj}^{2}+\sigma_{pej}^{2}\right)/\sigma_{Phj}^{2}$$

where 
$\sigma_{aj}^{2}$
 and 
$\sigma_{pej}^{2}$
 are additive genetic and permanent environmental variances at *DIM*
*j*; 
$z_{j}$
 is a vector of the Legendre polynomial coefficients evaluated at *DIM*
*j*; 
$h_{j}^{2}$
 is heritability of milk yield at *DIM*
*j*; 
$\sigma_{Phj}^{2}$
 is phenotypic variance obtained from sum of additive genetic variance, permanent environmental variance, and residual variance at *DIM*
*j*; and 
$r_{j}$
 is repeatability of milk yield at *DIM*
*j*.

### 2.5. Cross-Validation

The forward cross-validation method was used in the current study. This requires the definition of a cut-off date and construction of partial and whole data, which is based on the use of “old” and “recent + old” phenotype data, respectively [17,20]. In our study, the whole dataset was sliced after the first five years; it was the first partial data set that compromised records from 1990 to 1995. The whole data set was then sliced after the first six years and it was the second partial data set that contained records from 1990 to 1996. To simulate yearly genetic evaluations, this process continued until last year. In total, there were 21 consecutive evaluations with an interval of one year.

Since our objective was to select sires according to their daughters’ performance at an early age of lactation, we defined periods of early (E) and late (L) lactation, which contained cows with *DIM* in the range of 5–90 and 181–305, respectively, in each partial data set used in the cross-validation. To compare genetic evaluations, every two consecutive partial data sets were compared together as an old (*i* − 1) and new (*i*) partial data set. In each yearly evaluation, EBVs in the early period in the *i* − 1 partial data set were compared with EBVs in the late period in the *i* partial data set.

With the aim of comparing genetic evaluations, linear regression (LR) methods from [17] were used separately for all consecutive comparisons from *i* – 1 = 1, *i* = 2 until the last two partial data sets (*i* − 1 = 20, *i* = 21).

Let 
$\hat{u_{i}}$
 and 
$\hat{u_{i-1}}$
 be the EBV in the new (*i*) and old (*i* − 1) partial data sets.

(13)
$$b_{i,i-1}=cov\left(\hat{u_{i}},\hat{u_{i-1}}\right)/var\left(\hat{u_{i-1}}\right)$$

where 
$b_{i,i-1}$
 is the regression of EBV obtained with the new partial data set (*i*) on EBV estimated with the old partial data set (*i* − 1) and describes dispersion. The statistic 
$b_{i,i-1}$
 has an expected value of 1 if there is no over/under dispersion.

(14)
$${Þ}_{i,i-1}=cov\left(\hat{u_{i}},\hat{u_{i-1}}\right)/\sqrt{var\left(\hat{u_{i}}\right),\;var\left(\hat{u_{i-1}}\right)}$$

where 
${Þ}_{i,i-1}$
 is the correlation of EBV based on the old (*i* − 1) and new (*i*) partial data sets and shows population accuracy (correlation between TBV and EBV).

(15)
$$M_{i,i-1}=\bar{\hat{u_{i-1}}}-\bar{\hat{u_{i}}}$$

where 
$M_{i,i-1}$
 is deviation of average of EBV in the old (*i* − 1) and new (*i*) partial data sets and describes bias. The statistic 
$M_{i,i-1}$
 has an expected value of 0 if the evaluation is unbiased [17,18]. Figure 1 indicates cross-validation scenario.

In addition, the formulas below were used to estimate total phenotypic milk yields and then compute correlation between estimated total milk yield based on early lactation records and estimated total milk yield based on nearly completed lactations in the following yearly evaluation. This correlation was calculated to evaluate early selection, and cows in early lactation (5–90 days) in the *i* − 1 partial data set and that reached late lactation (181–305 days) in the *i* partial data set were involved.

(16)
TMY=EBVT+TPe


(17)
$${Þ}_{TMY\left(i,i-1\right)}=cov\left(\hat{TMY_{i}},\hat{TMY_{i-1}}\right)/\sqrt{var\left(\hat{TMY_{i}}\right),\;var\left(\hat{TMY_{i-1}}\right)}$$

where 
TMY
 is the estimated total milk yield; 
EBVT
 is the predicted total breeding value (from day 5 to 305); 
TPe
 is the sum of value of permanent environmental effect from day 5 to 305; and 
${Þ}_{TMY\left(i,i-1\right)}$
 is the correlation of estimated total milk yield based on the old (*i* − 1) and new (*i*) partial data sets.

## 3. Results

### 3.1. Fixed Effects

The solutions of fixed effects are illustrated in Figure 2. Figure 2A indicates yearly average solutions for herd-year effects from 1990 to 2015. The solutions for herds increased substantially over the whole period. Figure 2B shows solutions for age at calving as a contrast to age at calving at 50 months, which were set to zero. Milk yield increased from the age of 20 months and peaked around 26–28 months, whereafter yield gradually decreased as the cows become older at first calving. Figure 2C provides information about solutions for *DIM* in contrast to 305 *DIM*, which were set to zero. Milk yield production increased towards weeks 6–8 of the lactation period and then decreased. From day 275 onwards, estimates of milk yield production increased again.

### 3.2. Estimation of Genetic Parameters

Figure 3 provides information about genetic, permanent environmental, and phenotypic variances and their standard errors at different *DIM*. The permanent environmental variance was higher at the beginning and the end of the lactation period, and it had the highest value at 51 kg^2^ on the last day of the lactation period. The genetic variance was higher at the beginning of lactation, then decreased until the middle of lactation, and increased towards the end of lactation.

The heritability and repeatability of milk yield and their standard errors are indicated in Figure 4. The lowest heritability was estimated at just under 0.14 (±0.021) in early lactation, then increased dramatically towards the mid-lactation, and peaked at 0.18 (±0.024) on day 142 of lactation. The repeatability was high at the beginning of lactation, decreased towards mid-lactation, and bottomed out at 0.67 (±0.048) on day 154. After that, it increased again and was highest at 0.79 (±0.052) on day 305.

Genetic and phenotypic correlations between milk yield at different *DIM* are shown in Figure 5A,B, respectively. As can be seen in the graph, genetic correlations between adjacent test-days were high and declined with increasing distance between test-days. The genetic correlation between day 5 and days 6–13 was 0.99, between day 5 and days 58–83 was −0.82, and between day 5 and days 305–298 was −0.2.

### 3.3. Cross-Validation

The correlation of predicted breeding values in different yearly evaluations for sires and cows in E-L (early-late) periods is indicated in Figure 6. The E-L period includes cows with *DIM* in the range of 5–90 in the *i* − 1 partial data set and increased to *DIM* 181–305 in the *i* partial data set. Correlation for cows and sires was in the range of 0.77–0.87 and 0.81–0.94, respectively.

The results of 
$M_{i,i-1}$
 for cows and sires are indicated in Figure 7. In the E-L period, the statistic 
$M_{i,i-1}$
 was in the range of −72.71 to 498.70 and −81.74 to 155.84 for cows and sires, respectively. The statistic 
$M_{i,i-1}$
 had the highest value between the two last partial data sets (*i* − 1 = 20 and *i* = 21) for sires and cows. They were not significantly different from zero (*p* < 0.05) for deviation of average of estimated breeding values in *i –* 1 and *i* partial data set in all cow and sire groups.

The results of 
$b_{i,i-1}$
 and their standard errors are indicated in Figure 8. For cows and sires, the statistic 
$b_{i,i-1}$
 was in the range of 0.80–1.09 and 0.80–1.01, respectively. The statistic 
$b_{i,i-1}$
 describes dispersion, and when it is equal to 1, it means no over/under dispersion of predicted breeding values.

Figure 9 provides the results of 
${Þ}_{TMY\left(i,i-1\right)}$
. As shown in the graph, the correlation between estimated total milk yield in early and late lactation was in the range of 0.62–0.73 in different comparisons.

## 4. Discussion

### 4.1. Model

RR models are used for the genetic evaluation of milk yield trait because it allows studying the variation of trait as a function of time or stage of lactation. Therefore, finding the optimum order of fit in RR models is crucial. Several studies have indicated that the third order of fit is adequate for estimating variances because third-order polynomials follow the shape of genetic and permanent environmental variances over *DIM* with sufficient accuracy [24,25]. In the current study, various orders of polynomials were tested, and the optimum was selected based on likelihood ratio tests. In addition, the genetic correlation structure can be fitted with sufficient accuracy by a third-order polynomial [24,25,26]. Furthermore, a homogeneous residual variance was assumed through the lactation. The pattern of residual variance depends more upon the order of regression fitted to the permanent environmental effect than the regression fitted to the additive genetic effect. Thus, residual variance is not largely affected by the estimate of the additive genetic variance [27].

With the aim of avoiding assumptions on the form of the lactation curve that can be implied in polynomial regression, *DIM* and age at calving were modeled as class variables, as in several previous studies [15,24,26,28]. The solutions for the herd-year fixed effect increased by time, most likely due to improved management, including better diet and sanitation services. The solutions of age at calving peaked at 28 months and then reduced gradually towards a calving age of 49 months. In the fiftieth month, the effect of age at calving increased, and this peak is probably due to second-parity cows, where second calving is not recoded in our data. In first parity, cattle have higher milk production at the age of calving at 24–28 months, when they reach maturation age and gain proper weight for calving [28,29,30]. Solutions of *DIM* increased towards weeks 6–8 of the lactation period and then decreased toward the late lactation. This trend is expected for the early lactation period and is in agreement with other investigations [31,32]. The solution of *DIM* was raised again at the end of lactation, which can be due to missing calving dates in the data.

### 4.2. Genetic Parameters

The permanent environmental variance was higher in the beginning and toward the end of lactation, and that led to an increase in phenotypic variance and a decrease in heritability at the start and end of lactation. The first test-day is influenced by several environmental factors, such as feeding before calving, which results in larger variances at the beginning of lactation [13]. In addition, we had indications that the calving date was not always accurate in our data, which may increase variances in early lactation. Moreover, genetic variance can be changed over a generation because of selection [14]. However, we traced the pedigree to a base population to minimize the effect of previous selections. The heritability estimated in the present study was lower than the estimated values in other surveys that used RR models for Danish Holsteins, RR models for Iranian Holsteins, and multiple-trait linear models for Japanese Holsteins where the heritability of milk yield in their studies was estimated to be 0.36, 0.31, and 0.41, respectively [15,33,34]. In addition, the heritability of milk yield of Holstein-Friesian crossbred cattle and Tunisian dairy cattle using the Bayesian method was estimated to be 0.31 and 0.25, respectively [35,36]. Moreover, in other investigations where sire models were used for American Holsteins and bivariate repeatability animal models were applied for Italian Holsteins, the heritability of milk yield was estimated to be 0.12 and 0.11, respectively, in which the value of estimated heritability was lower than found in the present study [10,11]. The different estimates of heritability in various studies can be due to the number of animals, stage of lactation, feeding plan, and statistical methods [13]. In general, the highest heritability was observed in the middle of lactation, and lower heritability was observed in the beginning and end of lactation. This result is in agreement with several investigations where researchers used Bayesian analysis, RR models, and animal models [13,15,37]. The repeatability of milk yield had a higher value in the beginning and toward the end of lactation. Since repeatability is influenced by additive genetic and permanent environmental variances, it has a trend like the variances. The results of repeatability in the current study were in accordance with another study on Iranian Holsteins using RR models [38]. In another investigation on Holstein-Friesian dairy cattle using an animal model, repeatability was estimated in the range of 0.12–0.83 in different stages of lactation [12]. The estimated repeatability in our study was higher than in other surveys [39,40]. Moreover, the repeatability of milk yield on Tunisian dairy cattle using the Bayesian method was estimated to be 0.4, which was lower than our estimation [36]. Furthermore, the repeatability of milk yield estimated of Holstein-Friesian cattle using RR models was in the range of 0.80–0.91 in different stages of lactation, which was higher than estimated values in the current study [41]. Genetic correlations between adjacent test-days were high and declined with increasing distance between test-days. These results were in agreement with previous investigations completed using RR models [13,15,37,42,43,44].

### 4.3. Cross Validation

The correlation between predicted breeding values in *i −* 1 and *i* and averaged over all comparison partial data sets was high. In the E-L period that contained cows whose average *DIM* was in range of 5–90 in the *i* − 1 partial data set and reached to *DIM* 181–305 in the *i* partial data set, correlation for cows and sires was high. The EBVs of daughters in early lactation were highly correlated with the EBVs of daughters in late lactation, indicating that there is no need to wait until the end of the first lactation of the daughters to recognize superior sires. The statistic 
$M_{i,i-1}$
 had the highest value between the two last partial data sets for sires and cows, which can be because of fewer observations in the last year in our data. Nonetheless, in the E-L period the statistic 
$M_{i,i-1}$
 was not significantly different from 0 (*p* < 0.05) which means no detectable bias, and the statistic 
$b_{i,i-1}$
 was near 1, which indicates low over/under dispersion. On the other hand, early predicted breeding values in progeny groups was highly correlated to the final predicted breeding values. Moreover, the early prediction of total milk yield was positively correlated with the late prediction of total milk yield in *i* − 1 and *i* partial data sets which shows the reliability of early selection. Our results prove that we can select sires according to their daughters’ early lactation before finishing the first lactation and this early selection can be reliable. Early selection brings many benefits, such as decrease generation interval and increase genetic gain in the Iranian Holstein population.

## 5. Conclusions

The highest heritability of milk yield was observed in the middle of lactation and lower heritability in the beginning and end of lactation. Besides, genetic correlations between adjacent test-days were high and declined with increasing distance between test-days. 

Cross-validation showed a high positive correlation of predicted breeding values between consecutive yearly evaluations for both cows and sires. Moreover, early prediction of total milk yield was highly correlated with the late prediction of total milk yield. Consequently, sire selection can be completed based on their daughters’ early lactation performance. Early selection of sires results in a decreasing generation interval and speeding up of genetic gain, which is profitable for the dairy industry. Therefore, this early selection is suggested to breeders who do not have genomic information and want to recognize superior sires as soon as possible.

## Figures and Tables

**Figure 1 animals-11-03492-f001:**
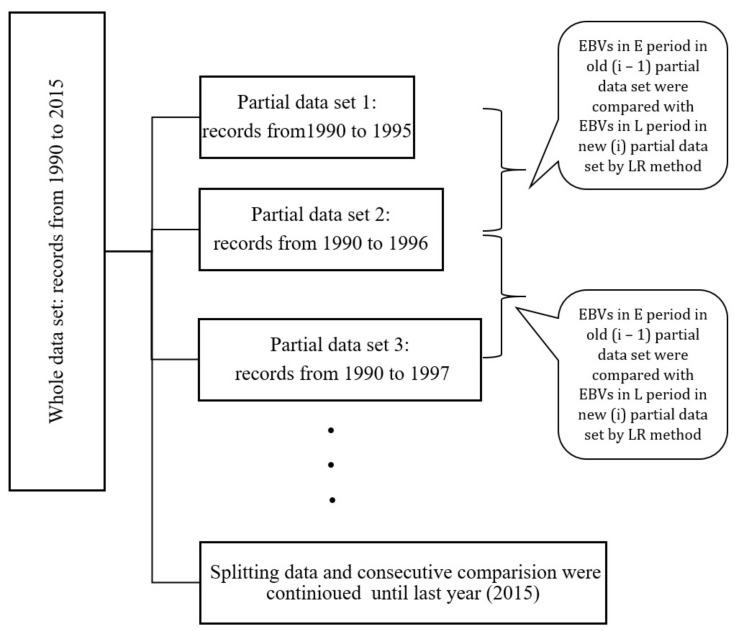
Forward cross-validation scenario used for emulating a system of yearly genetic evaluations. EBV: estimated breeding value, E: early period of lactation (5–90 days), L: late period of lactation (181–305 days), and LR: linear regression.

**Figure 2 animals-11-03492-f002:**
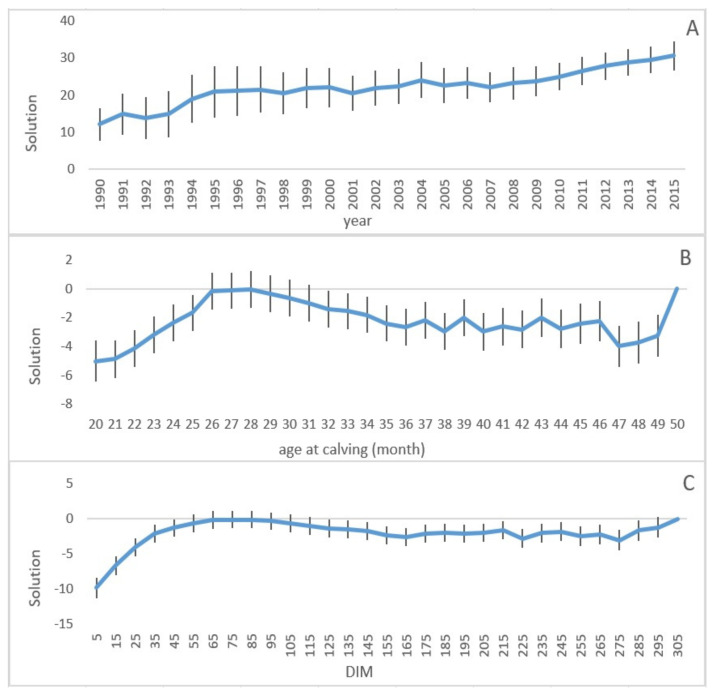
Solutions of fixed effects herd−year averaged over years (**A**), age of calving (**B**), and *DIM* (**C**); with standard error indicated. *DIM* = days in milk.

**Figure 3 animals-11-03492-f003:**
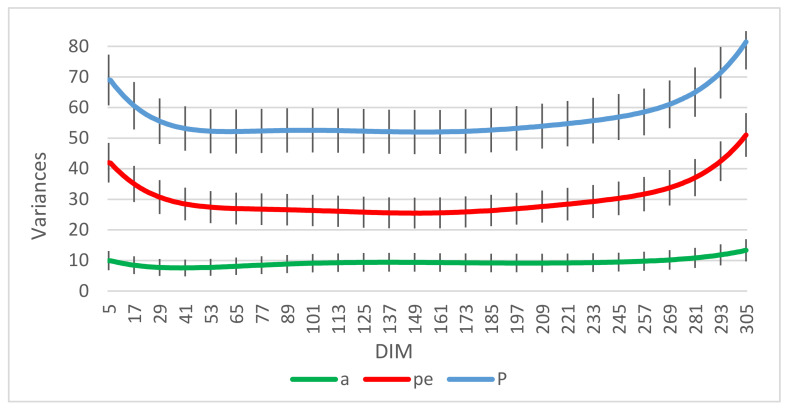
Genetic (a), permanent environmental (pe), and phenotypic (P) variances and their standard errors at different days in milk (*DIM*).

**Figure 4 animals-11-03492-f004:**
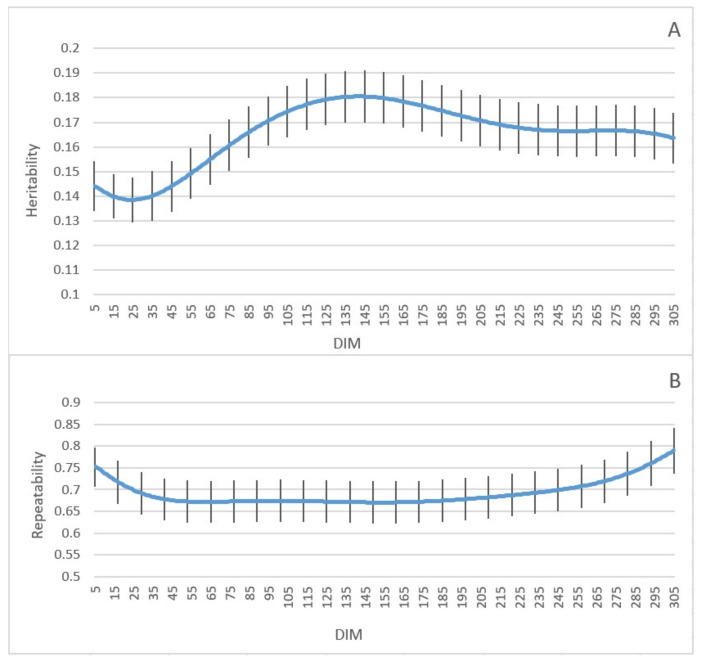
Heritability (**A**) and repeatability (**B**) of milk yield and their standard errors as a function of days in milk (*DIM*).

**Figure 5 animals-11-03492-f005:**
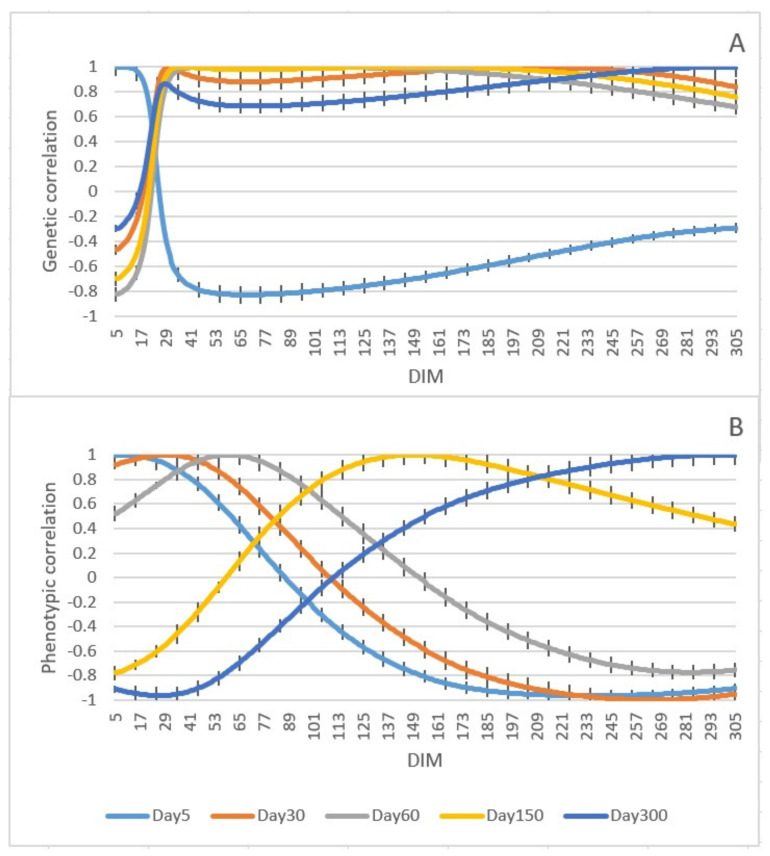
Genetic (**A**) and phenotypic (**B**) correlation between milk yields at different days in milk (*DIM*) with error bars indicating standard error.

**Figure 6 animals-11-03492-f006:**
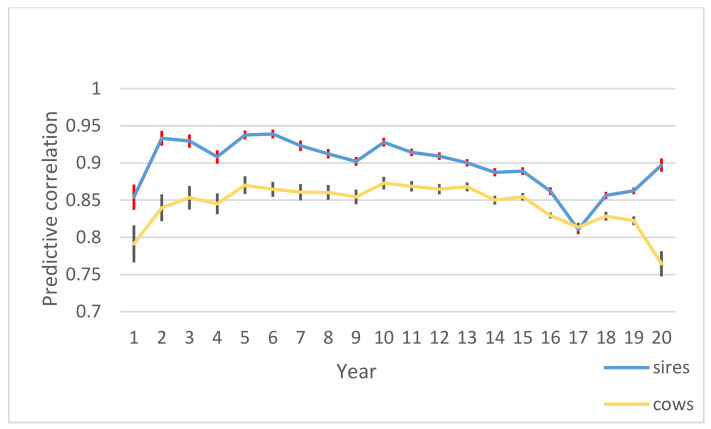
Correlation of estimated breeding values of cows and sires in E-L period. Years 1–20 are evaluated based on 1990–1995, followed by adding one year of data at a time.

**Figure 7 animals-11-03492-f007:**
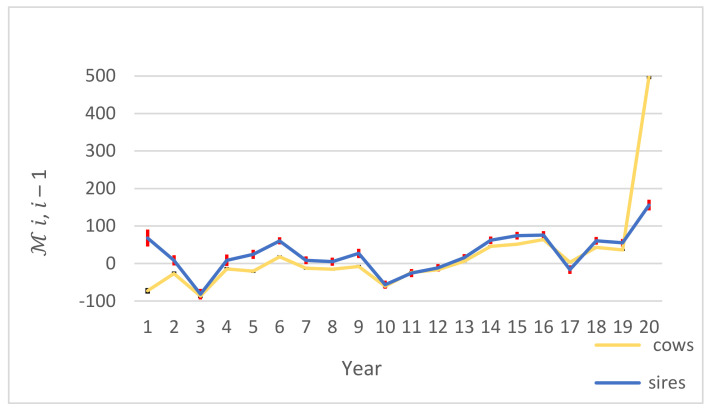
The deviation of average of estimated breeding values and their standard errors based on *i* and *i* − 1 partial data set in E-L period for cows and sires years 1–20 are evaluated based on 1990–1995, followed by adding one year of data at a time.

**Figure 8 animals-11-03492-f008:**
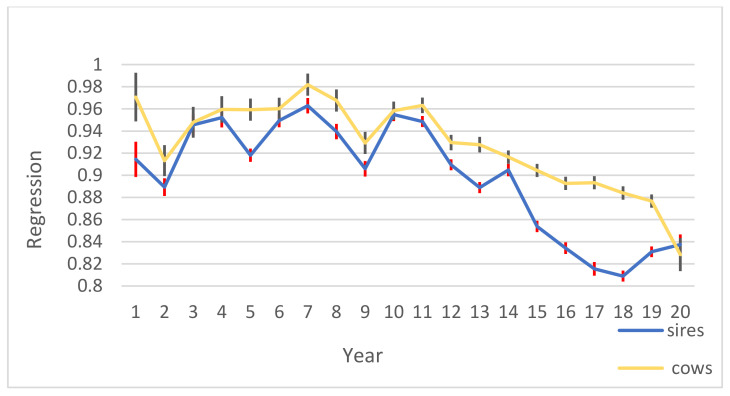
Regression of estimated breeding values of cows and sires and their standard errors in E-L period years 1–20 are evaluated based on 1990–1995, followed by adding one year of data at a time. Standard errors were in range of 0.004–0.022.

**Figure 9 animals-11-03492-f009:**
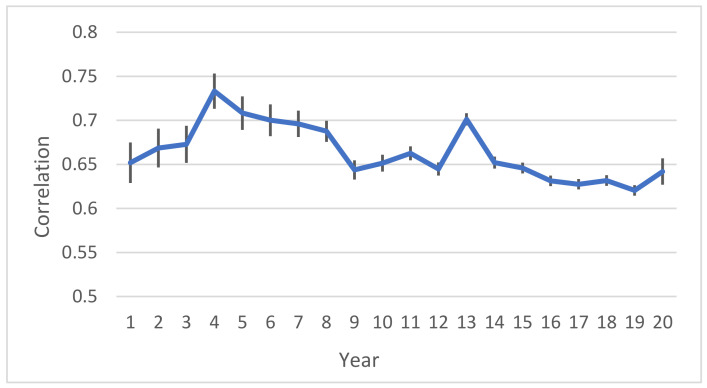
Correlation of estimated total milk yield in early and late lactation. Years 1–20 are evaluated based on 1990–1995, followed by adding one year of data at a time. Standard errors were in range of 0.006–0.023.

## Data Availability

Data set was obtained from the Animal Breeding Center of Iran. http://abc.org.ir (accessed on 7 December 2021).

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
