# Peer review of "Random Regression Model for Genetic Evaluation and Early Selection in the Iranian Holstein Population"

_animals, 2021, doi:10.3390/ani11123492_

Round 1

Reviewer 1 Report

Review for Salimiyekta et. al. manuscript.

General comments

In this paper, the authors present the “Random Regression model for genetic evaluation and early selection in Iranian Holstein population” for daily milk yield trait. using the deep pedigree information, which is of great interest in dairy cattle breeding programs. They used a Data consisted of 2,166,925 test-day records from 456,712 cows calving between 1990 and 2015. They have implemented a random regression model (RRM), with use of orthogonal Legendre polynomials. The goal of their research has been to prove that early lactation information from daughters may be used to appropriately select elite sires before they finish their first lactation.

Comments to the authors.

I consider that the manuscript represents a relevant contribution to the field of dairy farm breeding. The study presented here has some good results mainly meet the current milk production system and dairy cattle industry in that region. However, before being published in the "Animals" journal, the work requires a "major edit."

My general comments are as the follow:

  • The manuscript's written language and structure must be improved. For example, the introduction, contains a lot of general information regarding random regression models and cross-validation. As a result, I would advise the authors to be more specific and focus solely on the study aims, as well as clarify the problem and solutions they proposed in their work.

  • The results have not visualized very well, figure colors, legends and plot descriptions, all need to be extensively revised.

  • The cross-validation scheme that has been designed to assess the predictive performance of RR model not explained very well, and that is ambiguous for me.

Specific comments

Abstract

L1: Abstract needs to be revised in terms of reporting results and English.

L24: Average Information restricted maximum likelihood (AI-REML)

L25-26: A homogeneous residuls variance was assumed

L31-33- You need revise this sentence and report the findings in a more understandable way.

Introduction

L59-81: All of the sentences are unnecessary and provide general information; instead, you could discuss the problem in detail, as well as the shortage of your study inon this topic, and make a recommendation.

L82-85: I don't see any evidence that you developed a random regression model; instead, it appears that you took an already developed model and adjusted assumptions to fit your data. As a result, carefully reconsider your goals and objectives.

Materials and Methods

L90-91: All data edits were done in R environment (Please “R core team reference” ).

L104-107: This sentence must go at the end of your model description.  Instead, start this section by explaining the random regression model. For more information and instruction, I would suggest see several related papers.

L116-L117: The matrices in the equations should be in bold faces ( A, G0 and P0…).

L144: This sentence is general and should be removed or replace by defining the cross-validation schemes that you implemented.  Please see the following references for cross-validation scenarios.

Baba, T., Momen, M., Campbell, M.T., Walia, H. and Morota, G., 2020. Multi-trait random regression models increase genomic prediction accuracy for a temporal physiological trait derived from high-throughput phenotyping. PloS one15(2), p.e0228118.

Momen, M., Campbell, M.T., Walia, H. and Morota, G., 2019. Predicting longitudinal traits derived from high-throughput phenomics in contrasting environments using genomic Legendre polynomials and B-splines. G3: Genes, Genomes, Genetics9(10), pp.3369-3380.

L144-148. This explanation doesn't make sense to me. Please rewrite this section with more care and clarity, or create a diagram to show how you developed your cross validation scenario.

Results

Figure 1, plots A, B and C, please report only average of the milk yield over the time trajectory. You can use boxplot in R to have a better visualization.

L180: the “Test-day records were in range of 5-60 kg and average milk yield was 31.1±8.18 kg” is not clear for me? Average at what time point?

L195: Please start by explaining the Figure 2, here, for example: Figure 2 represents the estimated variance components for additive, permanent environment and … , using the random regression models explained above. The highest estimated values were obtained at … and ….

L198: “quadratic” and “cubic” effects, I don’t think they are effects, they are only “order of fit” for the Legendre Polynomial function.

L209-212: Please remove “to be”.

L208: Please re-write this section “The figure for 210 repeatability was ….”, you can explain the slope changes in heritability estimates over the trend.

L214: Please add more explanations for all Figure’s legends. What y-axis and x-axis represents? Do the same for Figure 4 and 5.

L221: In figure 4 A and B, both y-axes have the same label “correlation”.

L231: Please redraw this figure and make extra attempts to maintain the color of the SD bars consistent throughout all figures. Please relocate the "sire" and "dam" color legends to the right side or top of your plots, and the y-axis should be "Predictive correlation."

Discussion:

L257: This sentence is repetitive; please eliminate it or replace it with a sentence that is more specific. "From 1990 to 2015, our data consisted of milk production test-day records of first parity Iranian Holstein cows freshening."

L261: “Accuracy” please replace it by “predictive accuracy”.

L264-268: Please avoid repeating repeating words and statements.

L268: Finding the optimum order of fit in RR model is crucial, in this study we decided to choose the third order of fit according to the previous studies. …”

L277-279: This section is not clear for me! Please re-write it.

L333-338: Please remove this paragraph and avoid repeating words. I would suggest you to combine “Model”, “Genetic parameters”, and “Cross validation” sections in to one section and clean the discussion from repeated sentence and general statements.

Author Response

Dear reviewer

We appreciate your valuable comments and many thanks for your time and consideration. There is our answers to your comments.

General comments

In this paper, the authors present the “Random Regression model for genetic evaluation and early selection in Iranian Holstein population” for daily milk yield trait. using the deep pedigree information, which is of great interest in dairy cattle breeding programs. They used a Data consisted of 2,166,925 test-day records from 456,712 cows calving between 1990 and 2015. They have implemented a random regression model (RRM), with use of orthogonal Legendre polynomials. The goal of their research has been to prove that early lactation information from daughters may be used to appropriately select elite sires before they finish their first lactation.

 I consider that the manuscript represents a relevant contribution to the field of dairy farm breeding. The study presented here has some good results mainly meet the current milk production system and dairy cattle industry in that region. However, before being published in the "Animals" journal, the work requires a "major edit."

 My general comments are as the follow:

 The manuscript's written language and structure must be improved. For example, the introduction, contains a lot of general information regarding random regression models and cross-validation. As a result, I would advise the authors to be more specific and focus solely on the study aims, as well as clarify the problem and solutions they proposed in their work.

This manuscript was undergone English editing system.

 The results have not visualized very well, figure colors, legends and plot descriptions, all need to be extensively revised.

 The cross-validation scheme that has been designed to assess the predictive performance of RR model not explained very well, and that is ambiguous for me.

 Abstract

L1: Abstract needs to be revised in terms of reporting results and English. Revised

L24: Average Information restricted maximum likelihood (AI-REML). Revised

L25-26: A homogeneous residuls variance was assumed. Revised

L31-33- You need revise this sentence and report the findings in a more understandable way.

Revised

Introduction

L59-81: All of the sentences are unnecessary and provide general information; instead, you could discuss the problem in detail, as well as the shortage of your study inon this topic, and make a recommendation.

Revised and few words about background remained. This little background can lead to better understanding of M&M section. Besides, these few words can provide sufficient information for readers and give them an insight into RR and CV.

L82-85: I don't see any evidence that you developed a random regression model; instead, it appears that you took an already developed model and adjusted assumptions to fit your data. As a result, carefully reconsider your goals and objectives.

Revised

Materials and Methods

L90-91: All data edits were done in R environment (Please “R core team reference” ).

Revised

L104-107: This sentence must go at the end of your model description.  Instead, start this section by explaining the random regression model. For more information and instruction, I would suggest see several related papers.

Revised

L116-L117: The matrices in the equations should be in bold faces ( A, G0 and P0…).

Revised

L144: This sentence is general and should be removed or replace by defining the cross-validation schemes that you implemented.  Please see the following references for cross-validation scenarios.

 Revised

Baba, T., Momen, M., Campbell, M.T., Walia, H. and Morota, G., 2020. Multi-trait random regression models increase genomic prediction accuracy for a temporal physiological trait derived from high-throughput phenotyping. PloS one15(2), p.e0228118.

 Momen, M., Campbell, M.T., Walia, H. and Morota, G., 2019. Predicting longitudinal traits derived from high-throughput phenomics in contrasting environments using genomic Legendre polynomials and B-splines. G3: Genes, Genomes, Genetics9(10), pp.3369-3380.

 L144-148. This explanation doesn't make sense to me. Please rewrite this section with more care and clarity, or create a diagram to show how you developed your cross validation scenario.

 Revised

 Results

 Figure 1, plots A, B and C, please report only average of the milk yield over the time trajectory. You can use boxplot in R to have a better visualization.

Thanks for suggestion but Figure 1 shows the increase in yield over time and it indicates the effects of age at calving and days in milk. Besides, we discussed about these effects in the manuscript and this figure is essential to clear our discussion.

 L180: the “Test-day records were in range of 5-60 kg and average milk yield was 31.1±8.18 kg” is not clear for me? Average at what time point?

Revised

L195: Please start by explaining the Figure 2, here, for example: Figure 2 represents the estimated variance components for additive, permanent environment and … , using the random regression models explained above. The highest estimated values were obtained at … and ….

 Revised

L198: “quadratic” and “cubic” effects, I don’t think they are effects, they are only “order of fit” for the Legendre Polynomial function.

Revised

L209-212: Please remove “to be”. Revised

L208: Please re-write this section “The figure for 210 repeatability was ….”, you can explain the slope changes in heritability estimates over the trend.

 Revised

L214: Please add more explanations for all Figure’s legends. What y-axis and x-axis represents? Do the same for Figure 4 and 5.

Revised

 L221: In figure 4 A and B, both y-axes have the same label “correlation”.

Revised

L231: Please redraw this figure and make extra attempts to maintain the color of the SD bars consistent throughout all figures. Please relocate the "sire" and "dam" color legends to the right side or top of your plots, and the y-axis should be "Predictive correlation."

 Revised

Discussion:

 L257: This sentence is repetitive; please eliminate it or replace it with a sentence that is more specific. "From 1990 to 2015, our data consisted of milk production test-day records of first parity Iranian Holstein cows freshening."

Revised

L261: “Accuracy” please replace it by “predictive accuracy”.

Revised

L264-268: Please avoid repeating repeating words and statements.

Revised

L268: Finding the optimum order of fit in RR model is crucial, in this study we decided to choose the third order of fit according to the previous studies. …”

In our study, various orders of polynomials were tested and the optimum was selected based on likelihood ratio tests. This sentence has been mentioned in MM and Dis part.

L277-279: This section is not clear for me! Please re-write it.

 Revised

L333-338: Please remove this paragraph and avoid repeating words. I would suggest you to combine “Model”, “Genetic parameters”, and “Cross validation” sections in to one section and clean the discussion from repeated sentence and general statements.

Repeated sentences were deleted. Since in our result part we separated model, genetic parameters and CV, we prefer to have these parts in discussion section.

Yours sincerely,

Rasoul Vaez and Yasamin Salimi.

Reviewer 2 Report

Paper is interesting and it is always good to see new alternatives and science evolution in such a conservative field of quantitative genetics. However, a sufficient rationale must be provided to understand the validity of the alternatives or the better specificity of it for the case that this study approaches. Particularly liked cross validation section but, I need much more to be said on this. Authors speak of the goodness but not of the dark side or the drawbacks that using these method may have in the context of genetic evaluations.

Simple summary is well written and really depicts what is going to be performed. I would like to congratulate the authors because the achieved this in a very brief manner.

Abstract

The fact that the article deals with Iranian Holstein cows does not appear in any of the two summaries (simple or abstract) and should, given this provides the reader of a reference of something as important as the population evaluated in the study, which as the authors know is determinant for the estimation of genetic parameters, which are population specific.

If a model as the one referenced was used errors or credibility intervals must be supplied, for stats such as correlations or heritabilities.

Keywords: Try avoiding the use of words that were already in use in the title. Search engines already look for these, hence you miss an opportunity to be found.

Introduction.

References must follow the guides of the journal.

Line 45. Change EBV to regular fonts.

Line 47 to 50. The fact that pedigree testing is supported on SNP or microsatellite testing for parentage relationship and that such procedures improve predictive accuracy must be addressed.

Line 52 to 53. Please clarify. This point, if we consider the previous paragraph may be confounded with the fact that including ancestors increases reliability of predicted breeding values or estimated breeding values more than including new offspring in the matrix does, which is not totally true.

Line 56-57. Develop this idea, it is really important and I agree, but why does this happen?

Line 59 to 65. I agree, but BLUP, which has been the preferable option for years does this too. What is the rationale behind random regression being a rather feasible and reasonably more profitable option?

Line 67-68. I agree that if you understand cross validation methods what is said in the paper is correct. However, could you provide any reference to a work using the methodology? Computational problems may arise. For instance, to my mind comes that cross validation often involves subdividing the dataset into subsets (kfold) or removing observations (Loo). Doesn’t this change the genealogical/phenotypic structure of the population involved by definition? How can this be saved.

M&M

Lines 89 to 102.

Font is different.

I would rather have this information presented in a Table which describes all the pruning process with the inclusion or exclusion rationale and the final sample on which evaluations were carried.

To me, I am not sure if I would consider days in milk or age a fixed effect. It is true that the authors work with a huge sample but, this involves a huge range of levels. Indeed, this are normally considered in genetic evaluations as covariates which have their own correction coefficient within models and matrixes.

Line 152 to 153 and line 171 to 173. Please change font.

Cross validation seems to be correctly carried but the tools that were used are not described. If certain code or script was used this should be added as supplementary because the paper itself needs to be replicable. Anyway, I stick to my previous comment and potential distortion derived from data changes (due to subset creation or item removal) must be addressed.

Results

Lines 180 to 190. I still do not see ages being treated as fixed factor, even more when there representativity for each consecutive age or number of days in milk.

Bulmer effects may be becoming patent, as, after several generations of selection, the additive genetic variance and the rate of response to selection may become progressively asymptotic as kind of shown in Figures from 2 to 5. Could you discuss this?

Discussion

Line 257 to 263.

Could you please remove this? It has been repeated at other sections and it does not add anything to discussion. These are the objectives of the paper and should go in the last paragraph of introduction as they already are.

Discussion seems to be appropriate, but I would have love comparison with other methods such as BLUP and with other bovine populations to make it more internationally broad.

Conclusions are not conclusions. They repeat what have already been stated in other sections of the paper and are mostly results. What do we get out of these results? What is our take home message? Please use a single paragraph to create a simple extraction of the most relevant conclusions drawn from your results.

Author Response

Dear reviewer

We appreciate your valuable comments and many thanks for your time and consideration. There is our answers to your comments.

Paper is interesting and it is always good to see new alternatives and science evolution in such a conservative field of quantitative genetics. However, a sufficient rationale must be provided to understand the validity of the alternatives or the better specificity of it for the case that this study approaches. Particularly liked cross validation section but, I need much more to be said on this. Authors speak of the goodness but not of the dark side or the drawbacks that using these method may have in the context of genetic evaluations.

 Simple summary is well written and really depicts what is going to be performed. I would like to congratulate the authors because the achieved this in a very brief manner.

Abstract

The fact that the article deals with Iranian Holstein cows does not appear in any of the two summaries (simple or abstract) and should, given this provides the reader of a reference of something as important as the population evaluated in the study, which as the authors know is determinant for the estimation of genetic parameters, which are population specific.

Revised

If a model as the one referenced was used errors or credibility intervals must be supplied, for stats such as correlations or heritabilities.

Revised

Keywords: Try avoiding the use of words that were already in use in the title. Search engines already look for these, hence you miss an opportunity to be found.

Revised

Introduction.

References must follow the guides of the journal.

Revised

Line 45. Change EBV to regular fonts.

Revised

Line 47 to 50. The fact that pedigree testing is supported on SNP or microsatellite testing for parentage relationship and that such procedures improve predictive accuracy must be addressed.

Revised

Line 52 to 53. Please clarify. This point, if we consider the previous paragraph may be confounded with the fact that including ancestors increases reliability of predicted breeding values or estimated breeding values more than including new offspring in the matrix does, which is not totally true.

Revised

Line 56-57. Develop this idea, it is really important and I agree, but why does this happen?

Revised

Line 59 to 65. I agree, but BLUP, which has been the preferable option for years does this too. What is the rationale behind random regression being a rather feasible and reasonably more profitable option?

In many previous studies, animal and sire models were used for estimation of genetic parameters of milk yield. Whereas, milk yield production is significantly variable at different days in milk. Therefore, assumption of flexible genetic parameters across age is essential. RR allows studying variation of trait as a function of time and provide a better understanding of the genetics of lactation in dairy cattle

Line 67-68. I agree that if you understand cross validation methods what is said in the paper is correct. However, could you provide any reference to a work using the methodology? Computational problems may arise. For instance, to my mind comes that cross validation often involves subdividing the dataset into subsets (kfold) or removing observations (Loo). Doesn’t this change the genealogical/phenotypic structure of the population involved by definition? How can this be saved.

we developed our cross validation method to test the specific models and hypothesis raised in the paper. They are not standard CV but developed to test the hypothesis of the paper according to Leggara 2017. For all comparisons, we kept same pedigree to keep their structure.

M&M

Lines 89 to 102.

Font is different. Revised.

I would rather have this information presented in a Table which describes all the pruning process with the inclusion or exclusion rationale and the final sample on which evaluations were carried.

Thanks for your suggestion. I draw a table but the number of rows (or columns) were a lot and words in cells were very small and because of this, we preferred to keep this information in the text.

To me, I am not sure if I would consider days in milk or age a fixed effect. It is true that the authors work with a huge sample but, this involves a huge range of levels. Indeed, this are normally considered in genetic evaluations as covariates which have their own correction coefficient within models and matrixes.

We included dim as a class variable in order to avoid assumptions on the form of the lactation curve that can be implied in polynomial regression. In this way, there is no need for finding a function that fits well as a covariable. Test for lack of fit generally will show that the use of a class variable, which corresponds to a step function fits that data better than covariables.

Line 152 to 153 and line 171 to 173. Please change font.

Revised

Cross validation seems to be correctly carried but the tools that were used are not described. If certain code or script was used this should be added as supplementary because the paper itself needs to be replicable. Anyway, I stick to my previous comment and potential distortion derived from data changes (due to subset creation or item removal) must be addressed.

I just spitted full data set into 21 sub sets in R and then Ran each of them separately by DMU. There is no specific code for C.V. Besides, C.V part was revised and tried to clarify it.

Results

Lines 180 to 190. I still do not see ages being treated as fixed factor, even more when there representativity for each consecutive age or number of days in milk.

We included dim as a class variable in order to avoid assumptions on the form of the lactation curve that can be implied in polynomial regression. In this way, there is no need for finding a function that fits well as a covariable. Test for lack of fit generally will show that the use of a class variable, which corresponds to a step function fits that data better than covariables.

Bulmer effects may be becoming patent, as, after several generations of selection, the additive genetic variance and the rate of response to selection may become progressively asymptotic as kind of shown in Figures from 2 to 5. Could you discuss this?

Change in variances because of selection was added to discussion part.

Discussion

Line 257 to 263.

Could you please remove this? It has been repeated at other sections and it does not add anything to discussion. These are the objectives of the paper and should go in the last paragraph of introduction as they already are.

Revised

Discussion seems to be appropriate, but I would have love comparison with other methods such as BLUP and with other bovine populations to make it more internationally broad.

Revised. (References 11, 14) they used Bayesian method and BLUP.

Conclusions are not conclusions. They repeat what have already been stated in other sections of the paper and are mostly results. What do we get out of these results? What is our take home message? Please use a single paragraph to create a simple extraction of the most relevant conclusions drawn from your results.

Revised

Yours sincerely,

Rasoul Vaez and Yasamin Salimi.

Reviewer 3 Report

Dear authors,

The use of random regression model to estimate variance components and breeding values in Holstein cows is not new. However, since you used a quite big dataset, I think the manuscript can be considered for publication, but it must be modified. The manuscript is a little bit contorted, and it doesn’t read too well. Please try to remove some sentences that have been repeated too many times (see comments below). The paper can be simplified by removing some sentences in M&M (very hard to follow), results, and discussion that are just repetitions.

Introduction

Some “modern” findings are missing in the introduction, such as the use of genomic selection and changes in variance components over time; RR models can be used to study changes over time, but for example the time can be also divided into period to study these changes. Please see the comments and suggestions below.

Lines 47-49: please mention here that now genomic selection is replacing the traditional selection based on pedigree. Some examples

Lines 59-66: please mention here that variance components (and heritability) do not change only during the lifetime or physiological cycle of animals, but only during the time, mainly because of the genetic selection. Some recent references are: https://doi.org/10.1093/jas/skaa032; https://doi.org/10.1093/jas/skaa242; https://doi.org/10.3168/jds.2021-20151.

Lines 80-81: “Cross validation is a reliable…” this concept is already reported at lines 76-78.

Materials and Methods

Lines 93-95: these sentences are not useful.

Line 106: any citation for the “Orthopolynom” R package?

Line 111: did you use DIM as fixed effect or as linear covariate? Because it seems that you used the actual day of milk and then you had 305 levels, that are a lot to estimate. How many observations did you have for each DIM? This is useful to kwon also to interpret the plot of h2 across lactation: some artifacts are likely to appear at the beginning and end of lactation just because of the limited number of records.

Line 117: please clarify this sentence “? is relationship matrix of the order of number of animals”. Did you construct the relationship matrix using the pedigree? How many generations did you track back?

Line 145: please fix this sentence “Data set was cut off after first five years; it was the first data set (1990-1995).”

Line 154: “… EBV in early and late periods of same cows compared in …” should be “… were compared …” ?

Results

Line 180: “Test-day records were in range of 5-60 kg…” this is not a result; you cleaned the dataset to have this interval (as mentioned in the M&M section).

Line 192, Figure 1: the curve of DIM solutions is quite strange. Since DIM305 was set to zero, it is strange that solutions around the lactation pick are zero as at the end of lactation. Any comments on that?

Line 198-199: HTD and residual variances were fixed across the lactation, right? If so, please remove them from the plot. Please include standard deviation/error of these variances.

Line 214, Figure 3: as already mentioned, how many records did you have at the beginning of the lactation? The lower heritability can be an indicator of the lower number of records

Discussion

Lines 257-263: already mentioned several times.

Lines 265-269: these concepts have been repeated too many times in the paper.

Lines 275-279: these concepts are in contrast.

Lines 291-292: it can be because of the lower number of records that did not allow to have a correct estimate.

Line 328: “previous investigations that done by RR models” should be “previous investigations done using RR models”.

Conclusion

Line 357: please fix the conclusions paragraph. The first two sentences are more aim and results of the work. You should conclude the main outcomes of your research.

Author Response

Dear reviewer

We appreciate your valuable comments and many thanks for your time and consideration. There is our answers to your comments.

The use of random regression model to estimate variance components and breeding values in Holstein cows is not new. However, since you used a quite big dataset, I think the manuscript can be considered for publication, but it must be modified. The manuscript is a little bit contorted, and it doesn’t read too well. Please try to remove some sentences that have been repeated too many times (see comments below). The paper can be simplified by removing some sentences in M&M (very hard to follow), results, and discussion that are just repetitions.

 Introduction

Some “modern” findings are missing in the introduction, such as the use of genomic selection and changes in variance components over time; RR models can be used to study changes over time, but for example the time can be also divided into period to study these changes. Please see the comments and suggestions below.

Revised

Lines 47-49: please mention here that now genomic selection is replacing the traditional selection based on pedigree. Some examples

This statement was mentioned in another way in introduction “In more developed countries selection of dairy bulls is based on genotyped information from dense SNP arrays to predict genomic EBV using genomic prediction propecedures”

Lines 59-66: please mention here that variance components (and heritability) do not change only during the lifetime or physiological cycle of animals, but only during the time, mainly because of the genetic selection. Some recent references are: https://doi.org/10.1093/jas/skaa032; https://doi.org/10.1093/jas/skaa242; https://doi.org/10.3168/jds.2021-20151.

RR paragraph in introduction was revised and change in variances by time passing because of selection was mentioned in discussion section.

Revised

Lines 80-81: “Cross validation is a reliable…” this concept is already reported at lines 76-78.

 Revised

Materials and Methods

Lines 93-95: these sentences are not useful.

Revised

Line 106: any citation for the “Orthopolynom” R package?

Revised

Line 111: did you use DIM as fixed effect or as linear covariate? Because it seems that you used the actual day of milk and then you had 305 levels, that are a lot to estimate. How many observations did you have for each DIM? This is useful to kwon also to interpret the plot of h2 across lactation: some artifacts are likely to appear at the beginning and end of lactation just because of the limited number of records.

We included dim as a class variable in order to avoid assumptions on the form of the lactation curve that can be implied in polynomial regression. In this way, there is no need for finding a function that fits well as a covariable. Test for lack of fit generally will show that the use of a class variable, which corresponds to a step function fits that data better than covariables.

Line 117: please clarify this sentence “? is relationship matrix of the order of number of animals”. Revised

 Did you construct the relationship matrix using the pedigree? How many generations did you track back?

We traced back pedigree file by DMU-trace and did not put any restriction in parameter file. Thus, pedigree traced as far back as possible. There is histogram of traced generation for animal with records.

Line 145: please fix this sentence “Data set was cut off after first five years; it was the first data set (1990-1995).”

Revised

Line 154: “… EBV in early and late periods of same cows compared in …” should be “… were compared …” ?

Revised

Results

Line 180: “Test-day records were in range of 5-60 kg…” this is not a result; you cleaned the dataset to have this interval (as mentioned in the M&M section).

Revised

Line 192, Figure 1: the curve of DIM solutions is quite strange. Since DIM305 was set to zero, it is strange that solutions around the lactation pick are zero as at the end of lactation. Any comments on that?

Solution of DIM raised again at the end of lactation and it can be because of error in calving date, and cows were in second parity. (This sentence has mentioned in discussion part).

Line 198-199: HTD and residual variances were fixed across the lactation, right? If so, please remove them from the plot. Please include standard deviation/error of these variances.

Revised

Line 214, Figure 3: as already mentioned, how many records did you have at the beginning of the lactation? The lower heritability can be an indicator of the lower number of records

 Number of records at dim 5 were high (188265) then it decreased until day 22 and after that increased again.

Discussion

Lines 257-263: already mentioned several times.

Revised

Lines 265-269: these concepts have been repeated too many times in the paper.

Revised

Lines 275-279: these concepts are in contrast.

Revised

Lines 291-292: it can be because of the lower number of records that did not allow to have a correct estimate.

Number of records at dim 5 were high (188265) then it decreased until day 22 and after that increased again.

Line 328: “previous investigations that done by RR models” should be “previous investigations done using RR models”.

Revised

Conclusion

Line 357: please fix the conclusions paragraph. The first two sentences are more aim and results of the work. You should conclude the main outcomes of your research.

Revised

Yours sincerely,

Rasoul Vaez and Yasamin Salimi.

Round 2

Reviewer 1 Report

Dear Editor(s)

The authors have addressed my comment very effectively in the recent version, I only have two more comments and believe the work is of sufficient quality to be published.

  • In the figures 2, 3, 7 and 8, the legend for the y-axis should begin with a capital letter.
  • In equations 2-5, A, G0, P0 and Ip should be on bold face.

Kind regards

Author Response

Dear reviewer,

Many thanks for second revision and your time.

  • In the figures 2, 3, 7 and 8, the legend for the y-axis should begin with a capital letter.

Revised.

  • In equations 2-5, A, G0, P0 and Ip should be on bold face.
  • Revised.

Kind regards,

Rasoul Vaez & Yasamin Salimi.

Reviewer 2 Report

I think the authors have made an effort on applying still, I do not feel the categorization of age or dim is appropriate. The reason for this is that the values in the ends of each level are considered to have the same effect, which is often not correct. However, a less efficient approach is less efficient but not invalid, it is just that a deeper discussion must be supplied to express why the authors decided to do it so and which the drawbacks and benefits for it are.

In my opinion, this kind of approach adds too much to the error term of the equation, as the behaviour of this variables is not described by a function but by non natural categories prefixed by the researcher, but if researchers feel it is appropriate they need to provide evidences, rather than just say test for lack of fit generally will show that the use of a class variable, which corresponds to a step function fits that data better than covariables, because it mathematically sound as "yes you can evaluate what happen in each category separately" which si true, but not completely, indeed the intersections between categories and their information is missing, hence, the only manner in which we can figure out the behavior of the covariate is determining the function that it describes (that is the linear and nonlinear components within it, for milk production commonly they are linear and quadratic).

Author Response

Dear reviewer,

Many thanks for second revision and your time.

Line 177-181 were added regarding fixed effects.

Kind regards,

Rasoul Vaez & Yasamin Salimi.

Reviewer 3 Report

n/a

Author Response

Dear reviewer,

Many thanks for your time and consideration.

Line 50-51: new sentence was written and new reference was added.

Line 177-181 were added to explain more about cross-validation method.

Line 348-350 were added to explain about fixed effects.

Since you did not write specific comment, we were unable to alter the manuscript more. We hope these new sentences help the improvement of the manuscript.

Kind regards,

Rasoul Vaez & Yasamin Salimi.
